# Complex structures of Rsu1 and PINCH1 reveal a regulatory mechanism of the ILK/PINCH/Parvin complex for F-actin dynamics

Haibin Yang[1,2†], Leishu Lin[1†], Kang Sun[1], Ting Zhang[1,3], Wan Chen[1], Lianghui Li[1], Yuchen Xie[1], Chuanyue Wu[4]*, Zhiyi Wei[1]*, Cong Yu[1,5]*

[1]Department of Biology, Southern University of Science and Technology, Shenzhen, China; [2]Faculty of Health Sciences, University of Macau, Macau, China; [3]Academy for Advanced Interdisciplinary Studies, Southern University of Science and Technology, Shenzhen, China; [4]Department of Pathology, School of Medicine and University of Pittsburgh Cancer Institute, University of Pittsburgh, Pittsburgh, United States; [5]Guangdong Provincial Key Laboratory of Cell Microenvironment and Disease Research, and Shenzhen Key Laboratory of Cell Microenvironment, Shenzhen, China

**\*For correspondence:**
carywu@pitt.edu (CW);
weizy@sustech.edu.cn (ZW);
yuc@sustech.edu.cn (CY)

†These authors contributed equally to this work

**Competing interests:** The authors declare that no competing interests exist.

**Abstract** Communications between actin filaments and integrin-mediated focal adhesion (FA) are crucial for cell adhesion and migration. As a core platform to organize FA proteins, the tripartite ILK/PINCH/Parvin (IPP) complex interacts with actin filaments to regulate the cytoskeleton-FA crosstalk. Rsu1, a Ras suppressor, is enriched in FA through PINCH1 and plays important roles in regulating F-actin structures. Here, we solved crystal structures of the Rsu1/PINCH1 complex, in which the leucine-rich-repeats of Rsu1 form a solenoid structure to tightly associate with the C-terminal region of PINCH1. Further structural analysis uncovered that the interaction between Rsu1 and PINCH1 blocks the IPP-mediated F-actin bundling by disrupting the binding of PINCH1 to actin. Consistently, overexpressing Rsu1 in HeLa cells impairs stress fiber formation and cell spreading. Together, our findings demonstrated that Rsu1 is critical for tuning the communication between F-actin and FA by interacting with the IPP complex and negatively modulating the F-actin bundling.

## Introduction

F-actin cytoskeleton regulation is complicated and is critical for various cellular processes, including cell spreading, migration, proliferation, and apoptosis. Integrin-mediated focal adhesions (FAs) link the extracellular matrix (ECM) to intracellular integrin adhesion complex (IAC) and regulate F-actin dynamics (*Calderwood et al., 2000*; *Geiger et al., 2001*). Among the IAC components, tripartite integrin-linked kinase (ILK) associates with another two key players PINCH and Parvin to form the ILK/PINCH/Parvin (IPP) complex, which acts as an essential platform to recruit many proteins for the FA formation and signaling and provide a linkage between FA and actin cytoskeleton (*Legate et al., 2006*; *Qin and Wu, 2012*; *Tu et al., 2001*; *Wickström et al., 2010*; *Wu, 2004*). As a pseudokinase, ILK connects PINCH and Parvin through its N-terminal ankyrin repeat domain and C-terminal kinase domain, respectively (*Chiswell et al., 2008*; *Fukuda et al., 2009*; *Tu et al., 2001*; *Velyvis et al., 2001*). Dysfunctions of the three IPP proteins result in a number of diseases including different types of cancers, diabetes, and heart failure (*Cabodi et al., 2010*; *Hannigan et al., 2005*; *Qin and Wu,*

*2012*). Despite having been intensively investigated over the past two decades, the molecular mechanism underlying the function and regulation of the IPP complex remains elusive.

Ras suppressor protein 1 (Rsu1), a leucine-rich repeat (LRR)-containing protein conserved from human to worms (*Figure 1—figure supplement 1*), was recognized as a major FA component (*Horton et al., 2015*) and is important for FA formation and cell motility (*Simpson et al., 2008*; *Winograd-Katz et al., 2009*). Rsu1 was first identified to suppress Ras-dependent oncogenic transformation, during which the cells show typical anchorage-independent growth and morphological change (*Cutler et al., 1992*; *Masuelli and Cutler, 1996*). However, Rsu1 appears to possess multiple functions in cancer, as its expression level is upregulated in certain types of cancer cells and abnormally high expression level of Rsu1 may also contribute to cancer metastasis (*Gkretsi et al., 2017*; *Louca et al., 2020*; *Zacharia et al., 2017*).

Vertebrates express two PINCH proteins, PINCH1 and PINCH2, each consisting of a tandem array of five LIM domains (*Figure 1—figure supplement 2*). Rsu1 was reported to interact with the C-terminal LIM (LIM5) domain of PINCH1 (*Dougherty et al., 2005*; *Kadrmas et al., 2004*). By integrating Ras and integrin signaling, Rsu1 and PINCH1 concert to control cell adhesion, migration and apoptosis (*Dougherty et al., 2005*; *Dougherty et al., 2008*; *Kadrmas et al., 2004*; *Montanez et al., 2012*). Depletion of Rsu1 in cultured cells and *Drosophila* led to the reduced expression of PINCH1 and ILK, thus impairing the FA formation and the F-actin organization (*Dougherty et al., 2005*; *Gonzalez-Nieves et al., 2013*; *Kadrmas et al., 2004*). However, although Rsu1 was suggested to maintain the cellular protein level of the IPP complex that is important for the F-actin organization at the FA (*Legate et al., 2006*; *Wickström et al., 2010*), overexpressing Rsu1 damaged stress fibers, the cross-linked F-actin bundles anchored at the FAs in cells (*Masuelli and Cutler, 1996*). In addition, a recent study in *Drosophila* indicated that Rsu1 negatively regulates F-actin organization in muscle by inhibiting PINCH's activity (*Green et al., 2018*). Because the IPP complex was shown to promote F-actin bundling by using two actin-binding motifs from PINCH and Parvin respectively (*Vaynberg et al., 2018*), it is intriguing to unveil how Rsu1 coordinates its reversed effects in the destabilization of F-actin and the stabilization of the IPP complex.

Here, we solved the crystal structures of the Rsu1/PINCH1 complex and found that Rsu1 strongly binds to PINCH1 through a large concave surface on its LRR solenoid. The structural analysis revealed that the Rsu1-binding region and the actin-binding motif in PINCH1 are largely overlapped with each other. Consistently, the binding of Rsu1 to PINCH1 blocks the F-actin bundling activity of the IPP complex in vitro. In support of the inhibitory role of Rsu1 on the IPP-mediated actin-bundling, we overexpressed Rsu1 in HeLa cells and observed the impaired formation of stress fibers and the decreased cell spreading. Fusing the actin-binding motif in PINCH1 to Rsu1 eliminates the inhibitory effect of Rsu1, further validating that Rsu1 negatively regulates actin-bundling through specifically masking the actin-binding site on PINCH1. Together, our data demonstrated that Rsu1 through binding to PINCH1 modulates the actin-bundling function of the IPP complex for controlling FA dynamics and stress fiber formation for cell motility.

## Results

### Biochemical characterization of the interaction between Rsu1 and PINCH1

We investigated the binding of Rsu1 to PINCH1 by purifying various boundaries of both proteins (*Figure 1A,B*). The interaction between PINCH1 and RSU1 was confirmed by using isothermal titration calorimetry (ITC). The C-terminal region of PINCH1 containing either the LIM4 and LIM5 domains (LIM45C, aa.188–325) or the LIM5 domain (LIM5C, aa.249–325) shows the strong binding affinity to Rsu1 with a dissociation constant of ~15 nM (*Figure 1C,D*). Interestingly, the very C-terminal sequence (aa.308–325) in PINCH1 is indispensable for the binding of PINCH1 to Rsu1, as the LIM5 domain alone without this C-terminal sequence lost the Rsu1-binding ability (*Figure 1E*). On the other hand, the removal of the C-terminal region (aa.216–277) of Rsu1 (Rsu1ΔC) that does not belong to the LRR motif (*Kobe and Deisenhofer, 1994*) affected little to the complex formation of Rsu1 and PINCH1 (*Figure 1—figure supplement 3*), suggesting that the LRR-region is sufficient for the Rsu1/PINCH1 interaction.

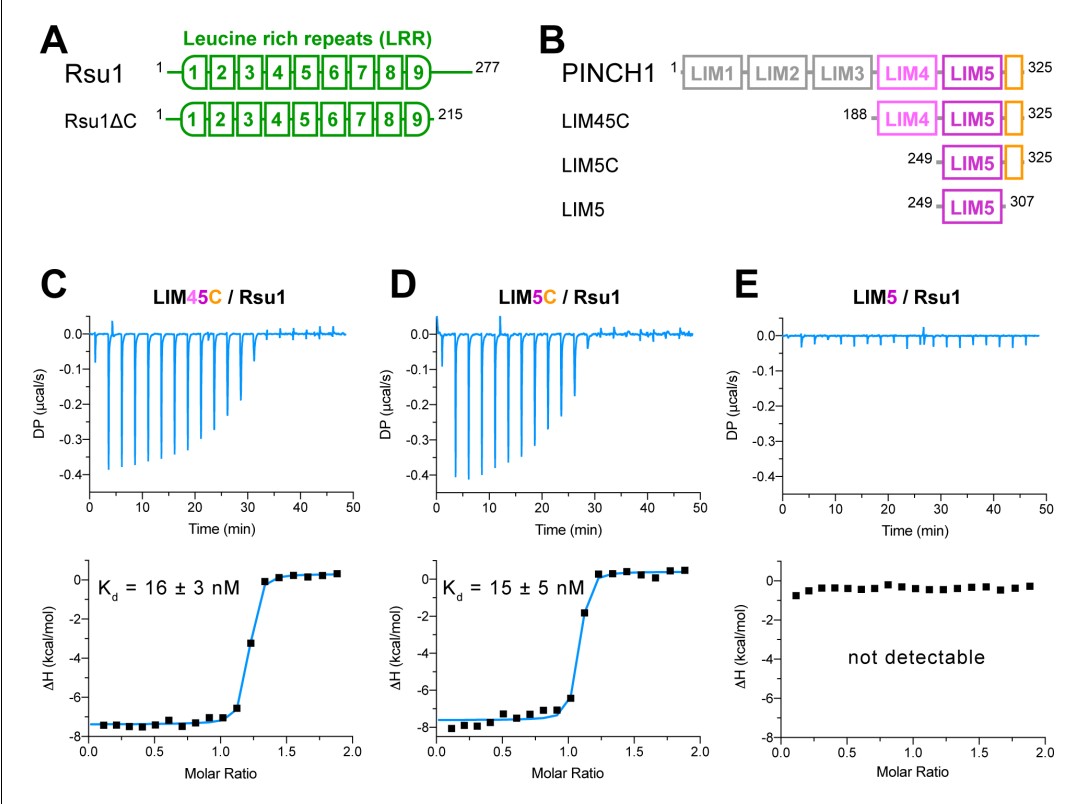

**Figure 1.** Biochemical characterization of the Rsu1/PINCH1 interaction. (**A** and **B**) Schematic domain organization of Rsu1 (**A**) and PINCH1 (**B**). The fragments used in this study are indicated. The color coding of the regions is applied in all figures as otherwise indicated. (**C–E**) Isothermal titration calorimetry (ITC) analysis of the interactions between Rsu1 and different PINCH1 boundaries.

The online version of this article includes the following figure supplement(s) for figure 1:

**Figure supplement 1.** Multiple sequence alignment of Rsu1 proteins from different species.

**Figure supplement 2.** Multiple sequence alignment of PINCH1_LIM45C from different species.

**Figure supplement 3.** Analytical gel filtration analysis of the interaction between the variants of Rsu1 and PINCH.

**Figure supplement 4.** Binding of LIM5C from PINCH2 or PINCH1 to Rsu1 analyzed by isothermal titration calorimetry (ITC).

## Overall structure of the Rsu1/PINCH1 complex

We prepared the Rsu1/PINCH1 complex by combinatorial usage of the purified fragments that contains the elements necessary and sufficient for the Rsu1/PINCH1 interaction and successfully obtained the crystals of the Rsu1ΔC/PINCH1_LIM5C and Rsu1/PINCH1_LIM45C complexes. The complex structures of Rsu1ΔC/PINCH1_LIM5C with one crystal form at 1.65 Å resolution and Rsu1/PINCH1_LIM45C with two different crystal forms at 2.2 Å and 3.15 Å resolution respectively were determined by the molecular replacement method (*Table 1*).

In the Rsu1/PINCH1_LIM45C complex, the nine LRRs of Rsu1 form a curved solenoid capped by the helical regions in the both N- and C-termini, named N-cap and C-cap, respectively (*Figure 2A,B* and *Figure 2—figure supplement 1A*). The N-cap consists of the α1-helix while the C-cap contains three α-helices (α2–4). Both the two helical regions cap the LRR-solenoid by forming extensive hydrophobic interactions with LRR1–2 and LRR8–9, respectively (*Figure 2C,D*). All the nine LRRs adopt the typical LRR-folding by using conserved leucine residues to form the hydrophobic core except for a few short loops inserted in LRR1, 5, and 9 (*Figure 2E,F*). Although lacking the C-terminal half of the LRR-motif, LRR9 remains folded presumably due to the presence of the C-cap. Consistently, the deletion of C-cap results in the disruption of LRR9 in the Rsu1ΔC/PINCH1_LIM5C complex (*Figure 2—figure supplement 1B*).

Consistent with our biochemical finding, the LRR-solenoid of Rsu1 interacts with both the LIM5 domain and the C-terminal tail (C-tail, folded as a α-helix) of PINCH1 through the concave surface of

**Table 1.** Statistics of data collection and model refinement.

| | Rsu1ΔC/PINCH1_LIM5 (PDB id: 7D2S) | Rsu1/PINCH1_LIM45 (7D2T/7D2U) | |
|---|---|---|---|
| **Data collection** | | | |
| Space group | I 4 | P 2₁ | I 2 2 2 |
| Cell dimensions | | | |
| a, b, c (Å) | 124.6, 124.6, 50.5 | 114.6, 51.3, 119.6 | 51.4, 144.4, 185.0 |
| α, β, γ (°) | 90, 90, 90 | 90, 101.6, 90 | 90, 90, 90 |
| Resolution (Å) | 50–1.65 (1.68–1.65) | 50–2.20 (2.24–2.20) | 50–3.15 (3.20–3.15) |
| $R_{merge}$* | 0.089 (0.931) | 0.152 (1.204) | 0.131 (0.967) |
| I/σI | 31.2 (2.8) | 14.1 (1.4) | 26.0 (1.9) |
| $CC_{1/2}^{†}$ | (0.823) | (0.687) | (0.885) |
| Completeness (%) | 100 (100) | 100 (100) | 99.9 (100) |
| Redundancy | 13.4 (12.9) | 6.7 (6.9) | 12.9 (12.4) |
| **Refinement** | | | |
| Resolution (Å) | 50–1.65 (1.69–1.65) | 50–2.20 (2.25–2.20) | 50–3.15 (3.44–3.15) |
| No. reflections | 46570 (2695) | 70228 (4880) | 12487 (3033) |
| $R_{work}/R_{free}^{‡}$ | 0.166 (0.241) / 0.185 (0.271) | 0.170 (0.281) / 0.198 (0.308) | 0.192 (0.262) / 0.216 (0.316) |
| No. atoms | | | |
| Protein | 2236 | 6055 | 2980 |
| Ligand/ion | 14 | 74 | 17 |
| Water | 187 | 358 | 0 |
| Mean B (Å) | | | |
| Protein | 32.4 | 53.1 | 134.2 |
| Ligand/ion | 32.2 | 78.1 | 151.2 |
| Water | 38.8 | 52.0 | - |
| r.m.s. deviations | | | |
| Bond lengths (Å) | 0.006 | 0.003 | 0.002 |
| Bond angles (°) | 1.01 | 0.74 | 0.55 |
| Ramachandran analysis | | | |
| Favored region (%) | 96.1 | 96.7 | 95.4 |
| Allowed region (%) | 3.9 | 3.3 | 4.6 |
| Outliers (%) | 0 | 0 | 0 |

The numbers in parentheses represent values for the highest resolution shell.

*$R_{merge} = \sum|I_i - I_m|/\sum I_i$, where $I_i$ is the intensity of the measured reflection and $I_m$ is the mean intensity of all symmetry related reflections.

†$CC_{1/2}$ is the correlation coefficient of the half data sets.

‡$R_{work} = \Sigma||F_{obs}| - |F_{calc}||/\Sigma|F_{obs}|$, where $F_{obs}$ and $F_{calc}$ are observed and calculated structure factors.

$R_{free} = \Sigma_T||F_{obs}| - |F_{calc}||/\Sigma_T|F_{obs}|$, where T is the test data set of about 4–5% of the total reflections randomly chosen and set aside prior to refinement.

the solenoid, mainly formed by the parallel β-sheet and the flanking loop regions (**Figure 2A,B,F**). In all three complex structures that we solved, the LIM5C fragment in PINCH1 and the LRRs in Rsu1 showed the essentially same conformation with the overall RMSD of ~0.3 Å (**Figure 2—figure supplement 1**). However, the LIM4 domain in PINCH1 adopts different orientations as observed in the two crystal forms of the Rsu1/PINCH1_LIM45C complex (**Figure 2—figure supplement 1A**), indicating the intrinsic, structural flexibility between the LIM4 and LIM5 domains.

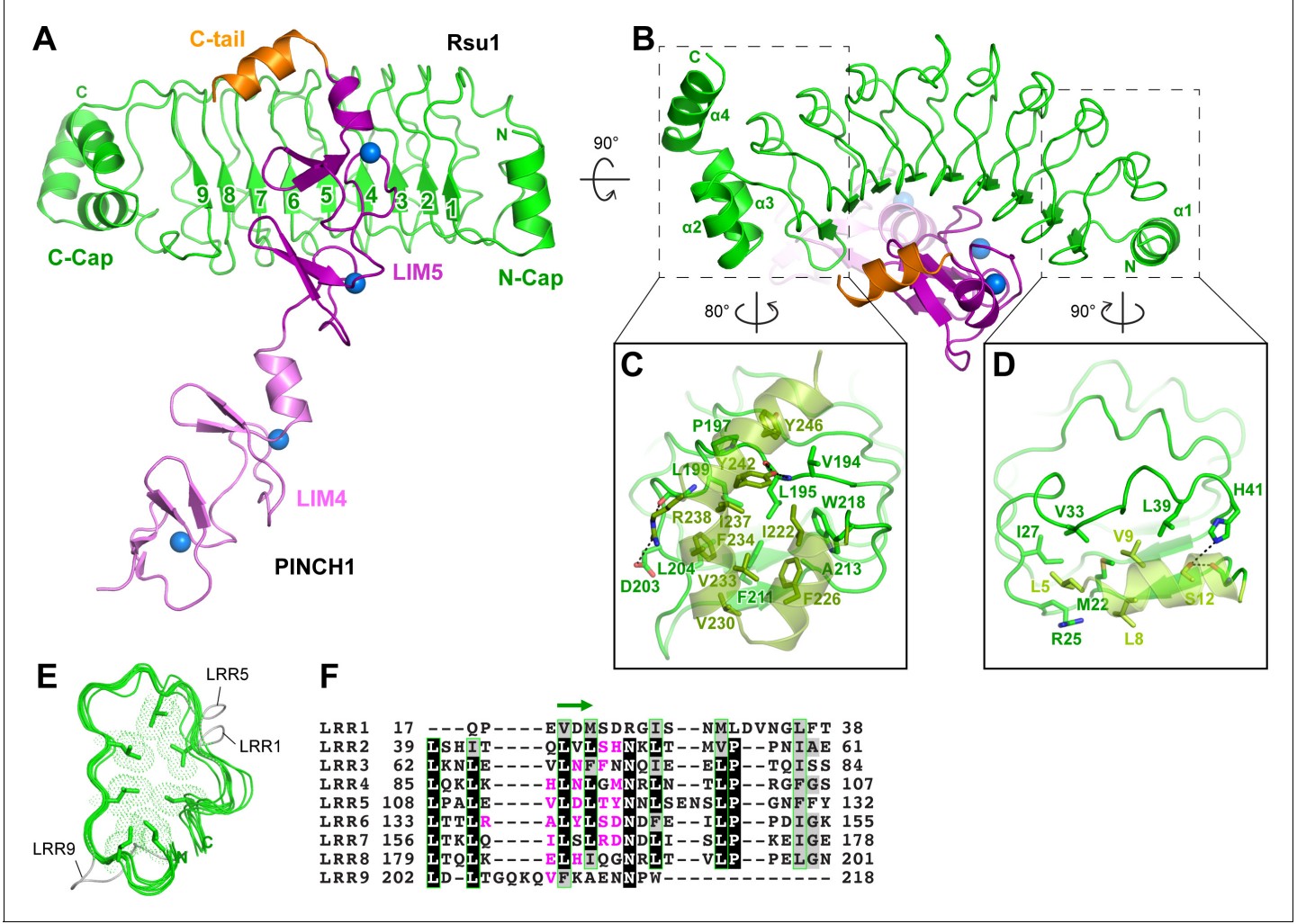

**Figure 2.** Structural analysis of the Rsu1/PINCH1_LIM45C complex. (A and B) Ribbon representations of Rsu1/PINCH1-LIM45C complex structure with two different views. Four $Zn^{2+}$ ions are indicated by blue spheres. (C and D) Molecular details of the C-terminal (C) and N-terminal (D) helices in Rsu1 capping the LRR-solenoid. Hydrogen bonds and salt bridges are indicated by dashed lines. (E) Structural alignment of the nine LRRs in Rsu1. The loops inserted in LRR1, LRR5, and LRR9 are indicated and the seven conserved hydrophobic residues that are mostly leucine and involved in forming the hydrophobic core of the LRR-solenoid are shown as sticks. (F) Sequence alignment of the nine LRRs in Rsu1. Identical and highly conserved residues are boxed in black and gray, respectively. The conserved hydrophobic residues are boxed in green and the amino acids involving in PINCH1 interaction are colored in magenta. The regions forming β-strands in the LRRs are indicated by a green arrow above the alignment.

The online version of this article includes the following figure supplement(s) for figure 2:

**Figure supplement 1.** Structural comparison of the Rsu1/PINCH1 complexes solved in this study.

## Molecular details of the Rsu1/PINCH1 interaction

In line with the nM-scale binding affinity, the Rsu1/PINCH1 interaction buries a large surface area of ~1100 Å$^2$ for each protein. The PINCH1-binding surface on Rsu1 is conserved from human to worm (*Figure 3A* and *Figure 1—figure supplement 1*). Given the highly conserved feature of the LIM45C region in all PINCH homologues (*Figure 1—figure supplement 2*), it is very likely that both PINCH1 and PINCH2 interacts with Rsu1 in different species using the same binding mode found in our structures. However, although the analytical gel filtration analysis showed that PINCH2_LIM5C forms a stable complex with Rsu1 in solution (*Figure 1—figure supplement 3D*), the binding of PIN-CH2_LIM5C to Rsu1 showed an affinity 100-fold weaker than the binding of PINCH1_LIM5C to Rsu1 (*Figure 1—figure supplement 4*), presumably due to the substitution of a few interface residues, such as F253 and H254 in PINCH1 replaced by a tyrosine and an asparagine in PINCH2, respectively. Upon binding to Rsu1, the C-terminal α-helix in PINCH1 undergoes a substantial conformational

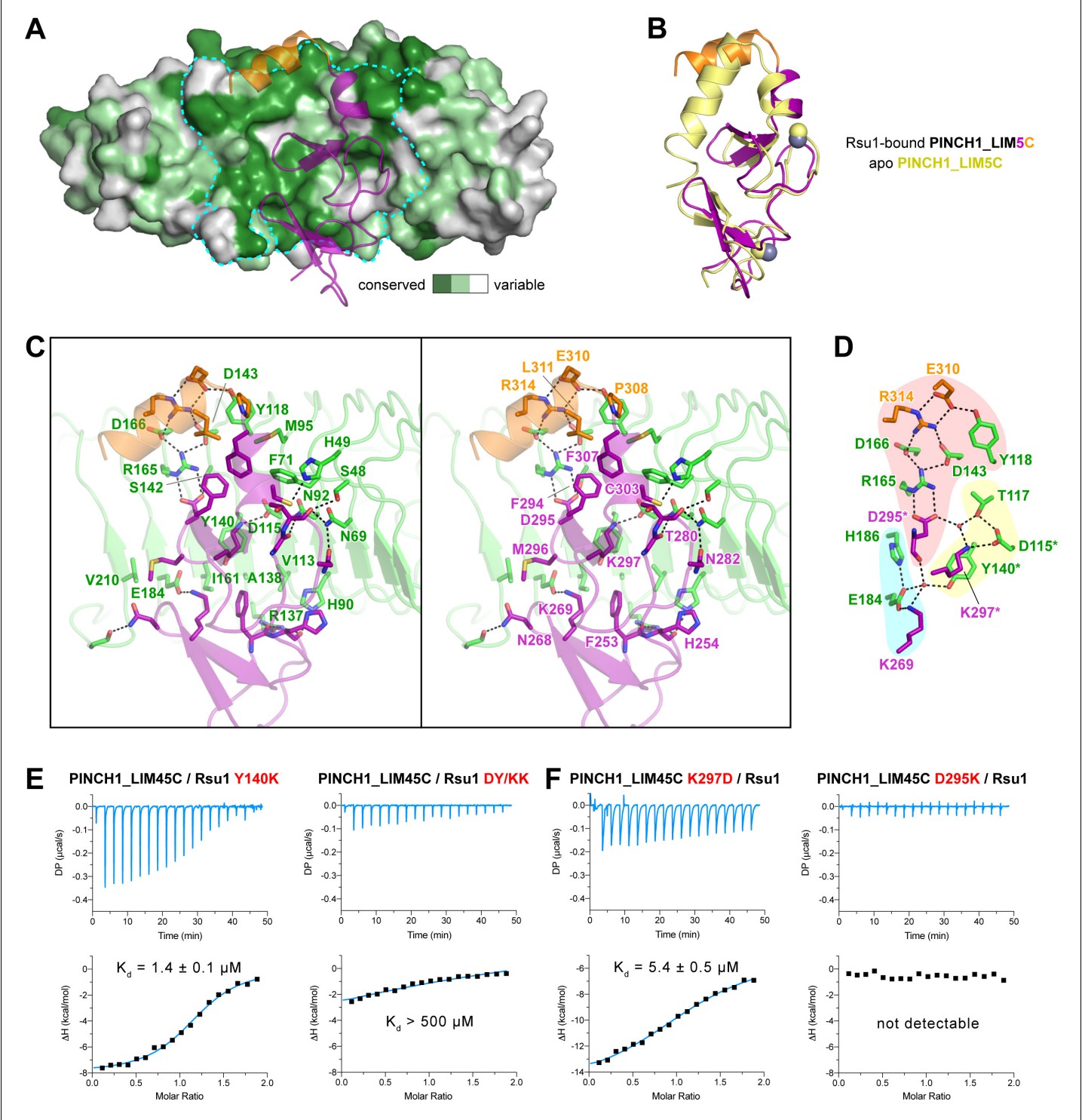

**Figure 3.** The Rsu1/PINCH1 interface. (**A**) Surface representations of Rsu1. The protein surface of Rsu1 is rendered with the amino acid conservation (*Figure 1—figure supplement 1*). (**B**) Structural superposition of the PINCH1_LIM5C structures in the apo (PDB id: 6MIF) and Rsu1-bounded forms. (**C**) Stereoview of the Rsu1/PINCH1 interface. Hydrogen bonds and salt bridges are indicated by dashed lines. (**D**) Polar interactions in the Rsu1/PINCH1 interface. These interactions are organized together through a hydrogen-bond network coordinately by three buried water molecules that are indicated as red balls. (**E and F**) Mutations of interfacial residues in either Rsu1 (**E**) or PINCH1 (**F**) impair the interaction between Rsu1 and PINCH1.

The online version of this article includes the following figure supplement(s) for figure 3:

**Figure supplement 1.** Electron density map of two buried water molecules in the Rsu1/PINCH1 complex structures.

*Figure 3 continued on next page*

*Figure 3 continued*

**Figure supplement 2.** Isothermal titration calorimetry (ITC)-based binding affinity measurements showing the impaired interaction between PINCH1 and Rsu1 with the interface mutations on Rsu1.

change as compared with the apo structure of PINCH1_LIM5C (*Figure 3B*; *Vaynberg et al., 2018*), while the conformation of the LIM5 domain remains largely unchanged.

The large PINCH1-binding surface on Rsu1 extends from LRR2 to LRR9 (*Figure 2F*), in which 22 surface residues form hydrogen bonds, salt bridges, and hydrophobic interactions with PINCH1_LIM5C (*Figure 3C*). These polar and hydrophobic interactions are interlaced with each other to provide the complementary binding required for the highly specific recognition between Rsu1 and PINCH1. Interestingly, although separately distributed at the interface, the polar interactions are well organized together by forming a hydrogen-bond network (*Figure 3D*). In this network, two buried water molecules, conserved in both the Rsu1ΔC/PINCH1_LIM5C and Rsu1/PINCH1_LIM45C complex structures (*Figure 3—figure supplement 1*), critically coordinate three clusters of the polar interactions as highlighted in *Figure 3D*.

Highly consistent with our structure analysis, interface mutations at either Rsu1 or PINCH1 disrupted the interaction between Rsu1 and PINCH1. In Rsu1, the Y140K mutation that disrupts the cation-π interaction between Y140$^{Rsu1}$ and K297$^{PINCH1}$, decreased the binding affinity by ~100 fold as indicated by ITC (*Figure 3E*), and the D115K/Y140K (DY/KK) double mutation further diminished the Rsu1/PINCH1 interaction (*Figure 3E* and *Figure 1—figure supplement 3E*) presumably due to the introduction of an additional charge repulsion in the binding interface. Likewise, the charge-reversing mutations (K297D and D295K) in PINCH1 either dramatically weakened or abolished the binding of PINCH1 to Rsu1 (*Figure 3E* and *Figure 1—figure supplement 3E*). In addition, several interface mutations of Rsu1 that disrupt the hydrogen-bonding, charge–charge interaction, or hydrophobic interaction between Rsu1 and PINCH1 showed decreased binding affinity (*Figure 3—figure supplement 2*).

## The PINCH1/Rsu1 interaction blocks the F-actin bundle formation promoted by the IPP complex

Previously, the C-terminal α-helix of PINCH1 has been shown to directly interact with actin and is required for the IPP-mediated bundling of F-actin, a process important for cell spreading and migration (*Vaynberg et al., 2018*). By comparing the Rsu1 and actin binding surfaces on PINCH1_LIM5C, we found that these two binding surfaces largely overlap with each other (*Figure 4A*), suggesting that the binding of Rsu1 to PINCH1 interferes with the binding of actin to PINCH1. Hence, we hypothesized that Rsu1 attenuates the IPP-mediated F-actin bundling by blocking the interaction between PINCH1 and actin (*Figure 4B*).

To test our hypothesis, we first purified the IPP complex (*Figure 4—figure supplement 1A*) and confirmed that Rsu1 binds to the IPP complex in a PINCH1-dependent way, as the wild-type Rsu1 but not the DY/KK mutant of Rsu1 forms a complex with IPP (*Figure 4C*). Next, we performed the actin bundling assay by adding either the purified IPP complex or the IPP/Rsu1 mixture to the polymerized actin, stained with Alexa488-phalloidin (*Vaynberg et al., 2018*). Consistent with our hypothesis, the actin-bundling ability of IPP complex was inhibited with the presence of Rsu1 in a concentration-dependent manner (*Figure 4D*). In contrast, the PINCH1-binding deficient mutants of Rsu1, Y140K, and DY/KK, even with a much higher concentration than wild-type Rsu1, were unable to interfere with the IPP-mediated F-actin bundling (*Figure 4D*). The detailed quantification further indicated that either the length or the number of bundled actins was decreased by the addition of wild-type Rsu1, but not the Y140K or DY/KK mutants (*Figure 4—figure supplement 2*). As a control, Rsu1 or its Y140K mutant could not bundle F-actin by itself, while the DY/KK mutant of Rsu1 weakly promotes F-actin bundling (*Figure 4—figure supplement 1B*), presumably induced by the artificial interaction between the introduced positively charged residues and actin. In addition, the electron microscopic analysis showed some large actin bundles in the F-actin sample with the IPP complex, but not in the F-actin sample with the IPP/Rsu1 mixture (*Figure 4—figure supplement 3*).

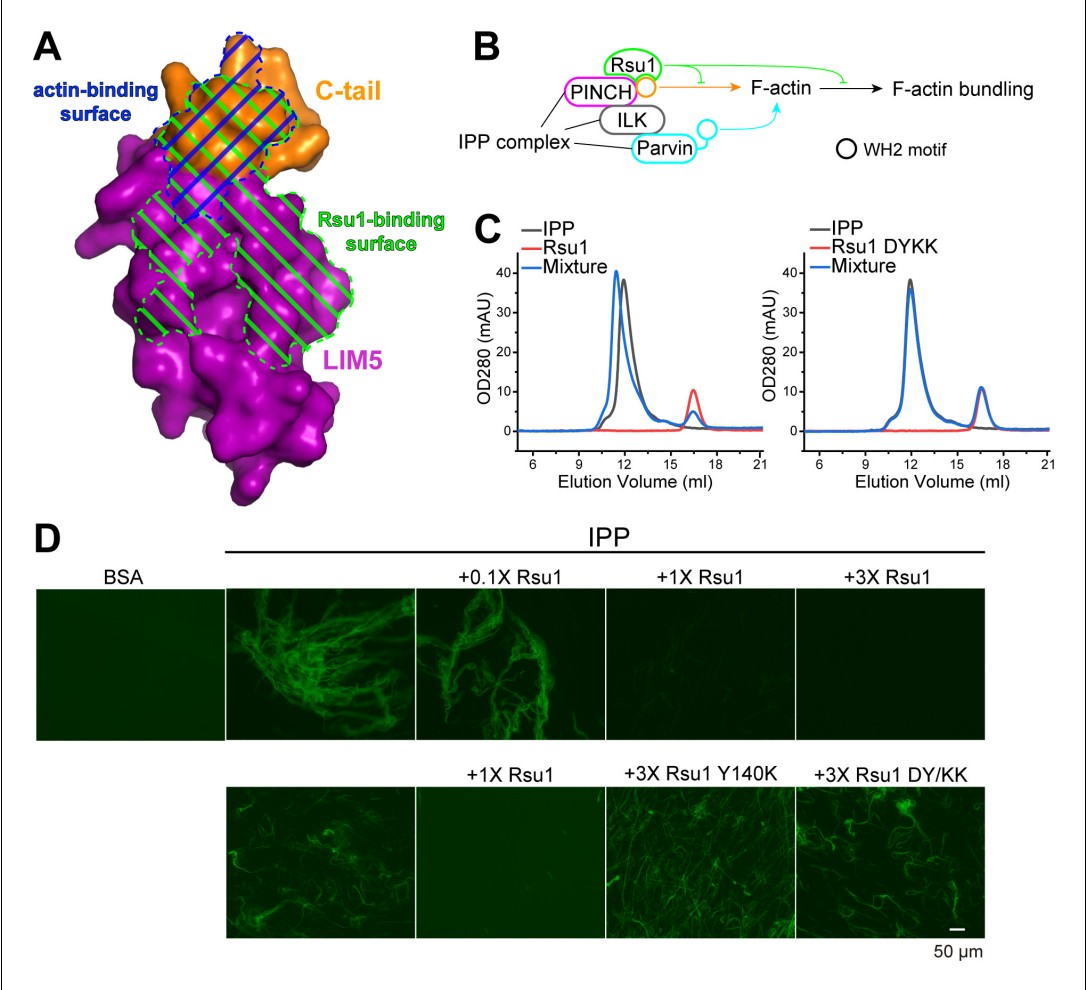

**Figure 4.** Rsu1 disrupts F-actin bundles induced by the ILK/PINCH/Parvin (IPP) complex. (**A**) Surfaces on PINCH1_LIM5C that are involved in the Rsu1-binding or actin-binding. (**B**) Schematic cartoon showing the inhibitory role of Rsu1 in the IPP-mediated F-actin bundling. (**C**) Rsu1 but not its DY/KK mutation can co-migrate with the IPP complex in size exclusion chromatography. The concentration for each protein is 20 µM. (**D**) The IPP-induced F-actin bundling is blocked by wild-type Rsu1 but not the PINCH1-binding defective mutations of Rsu1. The protein concentration of the IPP complex is 10 µM. 0.1×, 1×, and 3× indicate that the protein concentrations are 1 µM, 10 µM, and 30 µM, respectively.

The online version of this article includes the following figure supplement(s) for figure 4:

**Figure supplement 1.** Purification of ILK/PINCH/Parvin (IPP) complex and F-actin bundling assay of Rsu1 proteins.

**Figure supplement 2.** Quantification of actin bundles.

**Figure supplement 3.** Electron microscopic analysis of F-actin in the presence of ILK/PINCH/Parvin (IPP) or the IPP/Rsu1 mixture.

## The PINCH1/Rsu1 interaction regulates stress fibers and FA dynamics in cells

As previously reported, the IPP-induced F-actin bundling plays important roles in FA dynamics and stress fibers formation (*Vaynberg et al., 2018*). Since Rsu1 inhibits the F-actin bundling through the PINCH1 binding, it is likely that the interaction between Rsu1 and PINCH1 at the FA may regulate the FA-actin related cellular processes. To validate the cellular effect of Rsu1 on F-actin bundling, we overexpressed different Rsu1 variants in HeLa cells, the immortal human cervical cancer cells, and found that the wild-type Rsu1 localized to the FA while the PINCH1-binding deficient mutants (Y140K and DY/KK) failed to accumulate at the FA (*Figure 5A*). Since the stress fibers formation requires the IPP-mediated actin bundling (*Vaynberg et al., 2018*), we analyzed the cells transfected with Rsu1 or its mutants with a comparable expression level (*Figure 5—figure supplement 1*). Consistent with the inhibitory role of Rsu1 in the IPP-mediated actin bundling, stress fibers were significantly decreased in the cells transfected with Rsu1 while those remained largely unchanged in the

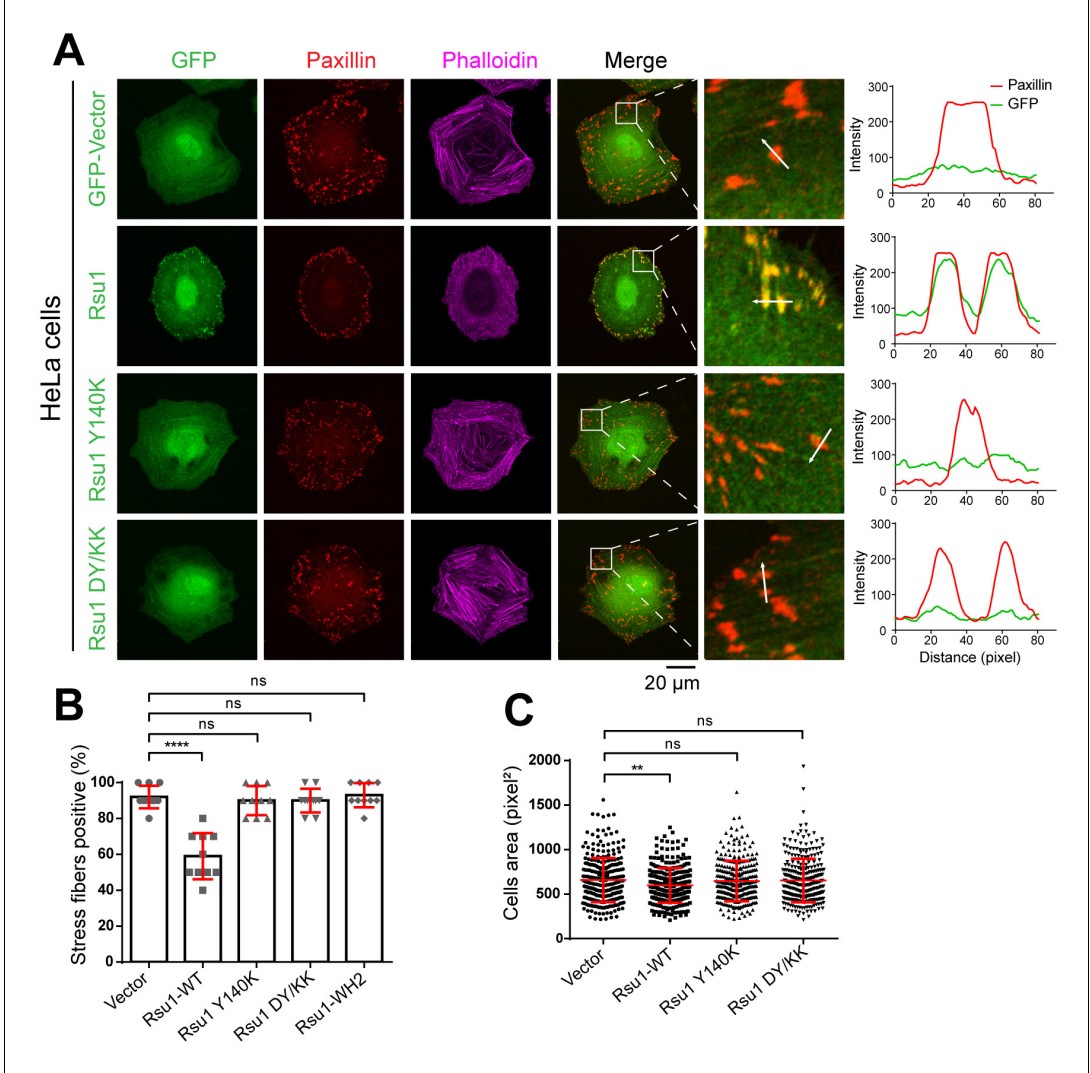

**Figure 5.** Overexpression of Rsu1 in HeLa cells regulates actin-related cellular processes. (A) Confocal images of HeLa cells transiently expressed of GFP, Rsu1-GFP, or Rsu1 mutations with a GFP tag. Focal adhesions (FAs) and stress fibers were stained by paxillin and phalloidin, respectively. (B) Quantification of the percentage of HeLa cells containing normal stress fiber formation as shown in **A**. Data was collected from 10 images in each sample. In each image, more than 10 cells were quantified. Significance was calculated by using Student's *t*-test in GraphPad Prism. ***p<0.001, *p<0.05. (C) Quantitative analysis of cell area in HeLa cells as imaged by Leica DMI6000B microscope. The areas of GFP-positive cells were measured using Image-Pro Plus. 300 cells were quantified in each sample. **p<0.01.

The online version of this article includes the following figure supplement(s) for figure 5:

**Figure supplement 1.** The expression level of wild-type Rsu1 and different mutants in HeLa cells as detected by western blot.

cells transfected with the Y140K and DY/KK mutants (*Figure 5A,B*). In addition, cell spreading was inhibited by overexpressing the wild-type Rsu1 (*Figure 5C*). In contrast, the overexpression of either Y140K or DY/KK in cells showed no decrease on cell spreading (*Figure 5C*). Thus, our cellular analysis supports that the binding of Rsu1 to PINCH1 attenuates the IPP's function on the actin-bundling.

## The inhibitory effect of Rsu1 is released by the fusion of the PINCH1-WH2 motif

The FA localization of Rsu1 largely relies on its binding to PINCH1 (*Figure 5A*). As the PINCH1-binding defective mutants of Rsu1 failed to accumulate at the FA (*Figure 5A*), these Rsu1 mutants may not interfere with the F-actin bundles at the FA. To further confirm the inhibitory effect of Rsu1 on

the IPP-mediated actin-bundling, we designed a Rsu1-WH2 chimaera, in which the WH2 motif of PINCH1 was fused to the C-terminus of Rsu1 (*Figure 6A*). Such a chimeric protein likely remains the PINCH1 association without inhibiting actin-bundling by providing another WH2 motif in the Rsu1/IPP complex (*Figure 6A*). In line with our hypothesis, Rsu1-WH2 bound to PINCH1_LIM45C with the nM-binding affinity, comparable to that of Rsu1 (*Figure 6B*). Meanwhile, contrasting to the inhibition of the actin-bundling by Rsu1, the IPP complex mixed with Rsu1-WH2 highly promoted the actin-bundling (*Figure 6C*). These results confirm that Rsu1 inhibits the IPP-mediated actin-bundling by masking the WH2 motif in PINCH1, as supplying the additional WH2 motif restores the decreased actin-bundling.

By overexpressing the Rsu1-WH2 chimaera in HeLa cells, we found that the chimeric protein accumulated at the FA (*Figure 6D*). To investigate whether the fused WH2 motif releases the inhibition of Rsu1 on actin bundling during the stress fiber formation, we quantitatively measured the numbers of the cells with normal stress fibers formation in HeLa cells transfected with either Rsu1 or Rsu1-WH2 with a comparable expression level (*Figure 5—figure supplement 1*). Compared with the decreased stress fiber formation in the cells overexpressing Rsu1, the cells overexpressing Rsu1-WH2 showed a normal level of stress fibers (*Figure 6D* and *Figure 5B*), suggesting that the inhibitory effect of Rsu1 on the actin bundles formation was eliminated by the fused WH2 motif.

## Discussion

Although Rsu1 was identified as the suppressor of Ras, the widely characterized oncogene, Rsu1 is upregulated in various cancers. The depletion of Rsu1 in the different cancer cells led to different influences on the cell motility (*Gkretsi and Bogdanos, 2015*; *Gkretsi et al., 2019*; *Louca et al., 2020*; *Simpson et al., 2008*), raising a more puzzling question about the Rsu1's role in regulating cell behaviors. In this study, we solved the complex structure of Rsu1 and PINCH1 and revealed that Rsu1 inhibits the IPP's function on actin bundling in vitro. The cellular data supported the inhibitory effect of Rsu1 on the IPP complex. These results indicated that Rsu1, by interacting with PINCH1 and preventing PINCH1 from binding to actin, modulates actin dynamics at the FA and thereby regulates the stress fiber formation and cell adhesion dynamics. Our findings provide a plausible mechanism to explain the possible oncogenic effect induced by the abnormally high protein level of Rsu1, which damaged the F-actin organization as we observed in cells. It is likely that cells tightly control the Rsu1 level for the homeostasis of actin dynamics, as either depleting Rsu1, which reduces the cellular protein level of the IPP complex and thereby impairs IPP-mediated actin bundling (*Kadrmas et al., 2004*; *Gonzalez-Nieves et al., 2013*), or elevating the Rsu1 level, which disrupts F-actin bundles, would interfere with the actin-dependent cell motility. In addition, Rsu1 binds to F-actin organizers, such as Rac1 and α-actinin (*Ojelade et al., 2015*; *Wang et al., 2020*), and regulates several downstream kinases of *Ras* oncogene, including JNK, ERK, and mitogen activated protein kinase 14 (p38) signaling (*Dougherty et al., 2008*; *Gonzalez-Nieves et al., 2013*; *Kim et al., 2019*; *Louca et al., 2020*; *Montanez et al., 2012*). Therefore, alterations of Rsu1 level may deregulate cell adhesion and migration and influence cancer cell behavior through multiple factors.

The high binding affinity between Rsu1 and PINCH1 ensures the inhibitory effect of even small amount of Rsu1 on the IPP-mediated actin organization. The nM-scale binding affinity is achieved through a large PINCH1-binding surface on the concave part of the LRR-solenoid in Rsu1 (*Figure 3A*), the most common protein interaction surface on the LRR proteins (*Kobe and Kajava, 2001*). Interestingly, two water molecules play important roles in mediating the Rsu1/PINCH1 interaction by integrating the hydrogen bond network that connects most polar residues in the interface (*Figure 3D*). Such a water-involved interaction network contributes to the binding energy. Disrupting the hydrogen bonding to one of the water molecules, like the Y140K or K297D mutation, cannot eliminate the binding of Rsu1 to PINCH1, as the interaction network may be partially maintained by the other water molecule. However, removing the linkages to both of the two water molecules by the DY/KK or D295K mutation destroyed the Rsu1/PINCH1 interaction.

The full-length structure of Rsu1 that we solved also provides a valuable information to uncover other binding partner(s) of Rsu1. Interestingly, in addition to the PINCH1-binding surface, we found an additional evolutionarily conserved surface on Rsu1 on the convex side of the LRR-solenoid (*Figure 6—figure supplement 1*), implicating that Rsu1 may interact with its binding partner via this surface. Notably, Rsu1 binds to the IPP complex with an affinity higher than those of other IPP-binding

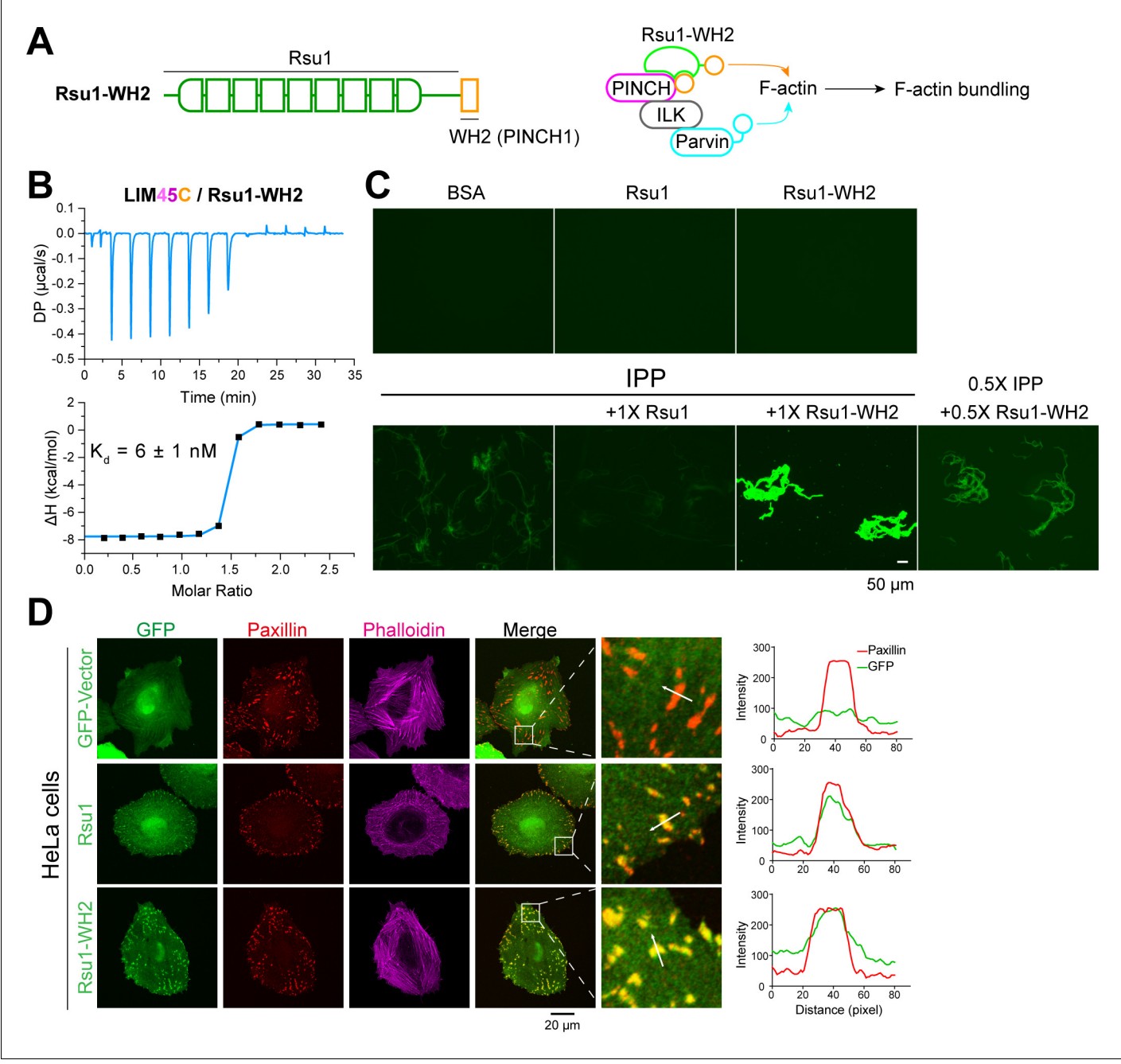

**Figure 6.** The inhibition of Rsu1 on the ILK/PINCH/Parvin (IPP) complex is released by the fusion of PINCH1-WH2. (**A**) Design of the Rsu1-WH2 chimera, in which the WH2 motif (308–325) of PINCH1 was fused to the C-terminus of Rsu1. This chimera provides an additional WH2 motif for F-actin bundling as indicated by the schematic cartoon in the right panel. (**B**) Isothermal titration calorimetry (ITC) analysis showing that Rsu1-WH2 binds to PINCH1 with an affinity comparable to wild-type Rsu1. (**C**) Actin bundling assay showing the bundling effect of the IPP complex in the presence of Rsu1-WH2. Unlike wild-type Rsu1, Rsu1-WH2 cannot block the IPP-mediated actin bundling. The 1× protein concentration is 10 µM. (**D**) Confocal images of HeLa cells transfected with Rsu1 or Rsu1-WH2. Similar to wild-type Rsu1, Rsu1-WH2 was localized at the focal adhesions (FAs). However, the formation of stress fibers was not impaired in the cells transfected with Rsu1-WH2.

The online version of this article includes the following figure supplement(s) for figure 6:

**Figure supplement 1.** Conservation analysis of the convex surface on Rsu1.

proteins (*Figure 1C*; *Fukuda et al., 2014*; *Stiegler et al., 2012*; *Vaynberg et al., 2005*). Considering that the IPP complex interacts with many actin-associated proteins (e.g. kindlin2, paxillin, and Thymosin-β4) at the FAs (*Bledzka et al., 2016*; *Legate et al., 2006*; *Wickström et al., 2010*), it is worthwhile to investigate the relationship between these proteins and Rsu1 in the regulation of the IPP-mediated actin organization and how these proteins and Rsu1 function together to regulate the integrin-actin crosstalk and consequently cell adhesion and migration.

## Materials and methods

### Cloning, protein expression, and purification

The human Rsu1 was expressed in insect cell expression system. The DNA sequence of Rsu1 or its mutants was amplified based on PCR and inserted into pFastBac HTB vectors. The constructs were transfected into DH10Bac competent cells for preparing Baculovirus genome (bacmid). The recombinant bacmid was then transfected into Sf9 for generating P1 virus and further amplified after two passages. The high tilter P3 virus was used to infect Sf9 cells for 72 hr. The infected cells were harvested and lysed by high-pressure homogenizers in the buffer (20 mM Tris-Cl pH 8.0, 500 mM NaCl, 5 mM imidazole, 1 mM PMSF freshly supplemented). The supernatant isolated from lysates by centrifuging at 20,000 rpm for 30 min was loaded to $Ni^{2+}$-NTA column. The eluted proteins were further purified by size-exclusion chromatography (GE Healthcare) with the buffer (50 mM Tris-Cl, pH 7.5, 100 mM NaCl, 2 mM DTT). The DNA sequence of human PINCH1_LIM45C (aa.188–325), LIM5C (aa.249–325) or mutants, and human PINCH2_LIM5C (aa.276–363) were amplified by PCR methods and inserted into pET32a vector with an N-terminal thioredoxin (Trx)-His for expression in BL21(DE3) *E. coli* cells. The purification procedure was essentially the same as used for Rsu1.

For preparing the complex samples, Rsu1 and PINCH1 protein were mixed at 1:1.5 molar ratio, incubation with TEV and PreScission protease to cut the tags, and further isolated by Superdex75 size exclusion column (GE Healthcare). For the expression of IPP complex, the pETDuet vector contained His-SUMO-tagged human ILK (residues 1–452) and His-tagged human α-parvin (residues 1–372) was co-transformed with the pRSF-SUMO vector inserted with human PINCH1(residues 1–325) in Rosetta (DE3). The cells were expressed in 16°C for ~18 hr and then were harvested for the purification. The complex protein was purified by $Ni^{2+}$-NTA column, followed by adding sumo protease to the elution to remove the SUMO-tag. The protein sample was further purified by size-exclusion chromatography (GE Healthcare) with the buffer (50 mM Tris-Cl, pH 7.5, 150 mM NaCl, 2 mM DTT). The IPP complex fractions after size-exclusion chromatography was verified by SDS-Page gel and collected. The proteins were refrigerated in −80°C for the following applications.

### Crystallization and data collection

Crystals of Rsu1/PINCH1 complex were obtained by the sitting drop vapor diffusion method in 16°C. To set up a sitting drop, 1 µl of concentrated protein solution (~25 mg/ml) was mixed with 1 µl of crystallization solution with 0.1 M sodium malonate pH 7.0, 12% w/v polyethylene glycol 3350 or 0.1 M Tris 8.0, 8% w/v PEG8000 for Rsu1/PINCH1_LIM45C, and 0.1 M Bicine pH 8.5, 20% w/v PEG10000 for Rsu1ΔC/PINCH1_LIM5C. Glycerol was gradually added into the crystal solution to 30% as the cryo-protectant before X-ray diffraction.

The diffraction data for structure determination was collected at Shanghai Synchrotron Radiation Facility beamlines BL17U1 (*Wang et al., 2018*), BL18U1, and BL19U1 (*Zhang et al., 2019*), and indexed and scaled by HKL2000 software package (*Otwinowski and Minor, 1997*). The structure of *Leptospira interrogans* LRR protein LIC10831 (PDB id: 4U06) was used as the search model in Phaser (*Storoni et al., 2004*) for phase determination of the Rsu1ΔC/PINCH1_LIM5C crystals by molecular replacement. LIM5C was manually built based on the electron density calculated by using the improved phase. The two structures of Rsu1/PINCH1_LIM45C were also solved by molecular replacement, which used the Rsu1ΔC/PINCH1_LIM5C structural model as the search model. These models were refined again in PHENIX (*Adams et al., 2010*). COOT was used for model rebuilding and adjustments (*Emsley and Cowtan, 2004*). In the final stage, an additional TLS refinement was performed in PHENIX. The model quality was check by MolProbity (*Davis et al., 2007*). The final refinement statistics are listed in *Table 1*. All structure figures were prepared by PyMOL (http://www.pymol.org/).

## Isothermal titration calorimeters

ITC experiments were performed on a PEAQ-ITC Microcal calorimeter (Malvern) at 25°C. Protein samples (in 50 mM Tris-Cl, pH 7.5, 100 mM NaCl) were prepared for titrations (13 or 19 titrations in total for each measurement). PINCH1_LIM45C or LIM5C and corresponding mutants with the concentration of 200 µM in the syringe were injected into the sample well containing 20 µM Rsu1 protein with the titration speed 0.5 µl/s. A time interval of 150 s between two titration points was applied to ensure that the titration peak returned to the baseline. The titration data were processed by MicroCal PEAQ-ITC Analysis Software and fitted by the one-site binding model.

## Analytical gel filtration chromatography and multi-angle static light scattering

Analytical gel filtration chromatography was carried out on an ÄKTA Pure system (GE Healthcare). Protein samples at indicated concentration were loaded onto a Superdex 200 Increase 10/300 GL column (GE Healthcare) equilibrated with 50 mM Tris-HCl buffer, pH 7.5, containing 100 mM NaCl.

The static light-scattering detector and differential refractive index detector (Wyatt) were coupled to the analytical gel filtration chromatography system. Data were analyzed with ASTRA6 provided by Wyatt.

## Actin bundling assay

Rabbit skeletal muscle globular actin (G-actin) was resuspended in GAB buffer (5 mM Tris-HCl pH 8.0, 0.2 mM $CaCl_2$, 0.5 mM DTT, 0.2 mM ATP) as prescribed (Cytoskeleton). G-actin concentration was determined by absorbance at 290 nm (with the extinction co-efficiency of 26,600). We polymerized 4 µM G-actin by addition of salts (50 mM KCl, 2 mM $MgCl_2$, 1 mM ATP) for half an hour at room temperature. Actin bundling assay was performed by mixing F-actin with tested proteins at indicated concentration and stayed in room temperature for 1 hr. Before observation by fluorescence microscopy, the mixture was labeled by AlexaFluor 488-Phalloidin (Thermo Fisher Scientific) and transferred to a coverslip. Images were taken by M2 upright microscope (Zeiss).

The quantification of the length and number of actin bundles was performed followed the previous description (*Breitsprecher et al., 2008*). Briefly, we polymerized 1 µM G-actin by the addition of salts (50 mM KCl, 2 mM $MgCl_2$, 1 mM ATP) for 10 min at room temperature and then mixed with buffer, 5 µM IPP protein, or the IPP/Rsu1 mixture. After labeling by AlexaFluor 488-Phalloidin, the samples were diluted 20 times with the GAB buffer plus salts (50 mM KCl, 2 mM $MgCl_2$, 1 mM ATP), and 5 µl samples were loaded to the coverslip for observation. Twenty images were taken for each sample by M2 upright microscope (Zeiss). The number and length of actin bundles were measured in ImageJ using Multi-point and Freehand Line tools.

## EM negative staining of F-actin

Actin bundles for TEM were prepared by mixing F-actin polymerized from 4 µM G-actin with 1 µM IPP or IPP/Rsu1, and stayed in room temperature for 1 hr. Mixture was dropped to glow discharged, carbon coated copper net (300 mesh), absorbing for 1 min, followed by negatively staining with 1% (w/v) uranium acetate for 30 s. Pictures were captured on TEM (HT7700, HITACHI).

## Cell lines

HeLa cell (serial number: TCHu187) was bought from National Collection of Authenticated Cell Cultures, Shanghai, China. The cells have been authenticated by STR profiling and have passed the mycoplasma contamination testing.

## Cell culture

HeLa cells were maintained in Dulbecco's modified eagle's medium supplemented with 10% fetal bovine serum and 1% penicillin–streptomycin solution. All cells were cultured at 37°C in incubator with 5% $CO_2$.

## Immunofluorescence and cell spreading assay

HeLa cells were transfected with pEGFP-N3-tagged constructs and GFP-positive cells were sorted by FACSAria Sorter (BD) after 24 hr to obtain the cells with a comparable expression level of GFP-

constructs, and then were transferred to fibronectin (12.5 µg/ml)-coated coverslips. After 2 hr, the cells were fixed with 4% paraformaldehyde (PFA, for 15 min at 37°C). After washing with PBS, the cells were treated with 0.1 Triton X-100 for 10 min at room temperature and blocked in 2% bovine serum albumin. The FAs were stained by anti-Paxillin antibody (Mouse, BD Bioscience), followed by Alexa Fluor 594 donkey anti-mouse secondary antibody (Thermo Fisher Scientific) and the F-actin was stained by Alexa Fluor 647 phalloidin (Thermo Fisher Scientific). The cells were visualized with $100\times$ objective using a Nikon A1R HD25 Confocal Microscope.

In cell spreading assays, the cells were stained by Alexa Fluor 594 phalloidin (Thermo Fisher Scientific) for the quantification of cell area, and the cells were visualized with $20\times$ objective using a Leica DMI6000B microscope. The cell areas of GFP-signaling positive cells were analyzed.

## Western blotting

Cells were lysed in lysis buffer (20 mM Tris, 100 mM KCl, 5 mM MgCl$_2$, 0.5% NP·40) and applied for protein quantification (Pierce TM BCA protein assay kit, Thermo Fisher Scientific). The lysates were mixed with loading buffer and subjected to SDS-PAGE. Proteins were then transferred to PVDF and blocked with 5% bovine serum albumin. The proteins were probed with primary antibodies (Rabbit anti-Rsu1, Thermo Fisher Scientific; Mouse anti-GFP, TRANS; Mouse anti-GAPDH, TRANS) and secondary antibodies (Anti-mouse HPR, Cell Signaling; Anti-mouse HPR, Cell Signaling).

## Statistics

Image pro plus was used to calculate the area of cells. The data was imported into Microsoft Excel and performed significance test of difference in GraphPad Prism software. Statistical significance was calculated by Student's $t$-test, and the differences were considered to be significant when p-value was less than 0.05. Line analysis of fluorescent intensity was conducted using Image pro plus and used GraphPad Prism for visualization.

## Acknowledgements

We thank Drs. Yi Deng, Ying Sun, Wenjie Wei, and Yilin Wang and Mr. Jie Liu for their suggestions on the experimental design. We thank the assistance of Southern University of Science and Technology (SUSTech) Core Research Facilities. We thank the staff from BL17U, BL18U, and BL19U1 beamlines of Shanghai Synchrotron Radiation Facility for assistance during data collection. This work was supported by the National Natural Science Foundation of China (Grant No. 31870757 to CY, 31970741 and 31770791 to ZW), Science and Technology Planning Project of Guangdong Province (2017B030301018), Shenzhen-Hong Kong Institute of Brain Science, and Shenzhen Fundamental Research Institutions (2019SHIBS0002). ZW is a member of the Brain Research Center, SUSTech.

## Additional information

### Funding

| Funder | Grant reference number | Author |
|---|---|---|
| National Natural Science Foundation of China | 31870757 | Cong Yu |
| National Natural Science Foundation of China | 31970741 | Zhiyi Wei |
| National Natural Science Foundation of China | 31770791 | Zhiyi Wei |
| Science and Technology Planning Project of Guangdong Province | 2017B030301018 | Cong Yu |
| Shenzhen-Hong Kong Institute of Brain Science, Shenzhen Fundamental Research Institutions | 2021SHIBS0002 | Zhiyi Wei |

The funders had no role in study design, data collection and interpretation, or the decision to submit the work for publication.

## Author contributions
Haibin Yang, Designed constructs, Purified proteins and performed biochemistry assays, Assisted with preparing the manuscript; Leishu Lin, Performed all the cellular assays, Assisted with preparing the manuscript; Kang Sun, Ting Zhang, Assisted with designing and performing cellular assays; Wan Chen, Assisted with designing constructs and pufiringproteins; Lianghui Li, Yuchen Xie, Assisted with pufiring proteins; Chuanyue Wu, Supervision, Writing - review and editing; Zhiyi Wei, Supervision, Funding acquisition, Writing - original draft, Writing - review and editing, Solved and analyszed structures; Cong Yu, Conceptualization, Supervision, Funding acquisition, Writing - original draft, Project administration, Writing - review and editing, Aesigned all the experiments

## Author ORCIDs
Haibin Yang ![ORCID] http://orcid.org/0000-0002-6902-6941
Leishu Lin ![ORCID] https://orcid.org/0000-0002-1214-4760
Zhiyi Wei ![ORCID] https://orcid.org/0000-0002-4446-6502
Cong Yu ![ORCID] https://orcid.org/0000-0003-2912-6347

## Decision letter and Author response
Decision letter https://doi.org/10.7554/eLife.64395.sa1
Author response https://doi.org/10.7554/eLife.64395.sa2

# Additional files
## Supplementary files
• Transparent reporting form

## Data availability
Diffraction data have been deposited in PDB under the accession code 7D2S, 7D2T and 7D2U.

The following datasets were generated:

| Author(s) | Year | Dataset title | Dataset URL | Database and Identifier |
|---|---|---|---|---|
| Yang H, Wei Z, Cong Y | 2021 | Crystal structure of Rsu1/PINCH1_LIM5C complex | https://www.rcsb.org/structure/7D2S | RCSB Protein Data Bank, 7D2S |
| Yang H, Wei Z, Cong Y | 2021 | Crystal structure of Rsu1/PINCH1_LIM45C complex | https://www.rcsb.org/structure/7D2T | RCSB Protein Data Bank, 7D2T |
| Yang H, Wei Z, Cong Y | 2021 | Crystal structure of Rsu1/PINCH1_LIM45C complex | https://www.rcsb.org/structure/7D2U | RCSB Protein Data Bank, 7D2U |

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

# Appendix 1

**Appendix 1—key resources table**

| Reagent type (species) or resource | Designation | Source or reference | Identifiers | Additional information |
|---|---|---|---|---|
| Gene (*Homo sapiens*) | RSU1 | GenBank | NP_036557.1 | |
| Gene (*Homo sapiens*) | PINCH1 | GenBank | NP_001180417.1 | |
| Gene (*Homo sapiens*) | PINCH2 | GenBank | NP_001154875.1 | |
| Gene (*Homo sapiens*) | ILK | GenBank | NP_001014794.1 | |
| Gene (*Homo sapiens*) | Parvinα | GenBank | NP_060692.3 | |
| Recombinant DNA reagent | pFastBac-HTB-RSU1 | This paper | (1-277) / Full-length | BamHI/XhoI |
| Recombinant DNA reagent | pFastBac-HTB-RSU1 | This paper | (1-215) / LRR | BamHI/XhoI |
| Recombinant DNA reagent | pFastBac-HTB-RSU1 | This paper | /N69A/F71Q/R137E/Y140K/D143K/ R165A/D115K-Y140K/D115K-D143K/ F71Q-D115K-Y140K/WH2(308–325)/ | Kinds of Mutations in full-length gene |
| Recombinant DNA reagent | pEGFP-N3-hRSU1 | This paper | (1-277) / Full-length | HindIII / BamHI |
| Recombinant DNA reagent | pEGFP-N3-hRSU1 | This paper | /Y140K/D115K-Y140K/WH2/ | Kinds of Mutations in full-length gene |
| Recombinant DNA reagent | pET.32M.3C | PMID:19665975 | | Dr. Mingjie Zhang (SUSTech, China) |
| Recombinant DNA reagent | pET.32M.3C-hPINCH1 | This paper | LIM5C(249-325) | BamHI/XhoI |
| Recombinant DNA reagent | pET.32M.3C-hPINCH1 | This paper | LIM45C(188-325) | BamHI/XhoI |
| Recombinant DNA reagent | pET.32M.3C-hPINCH1 | This paper | LIM45(188-307) | BamHI/XhoI |
| Recombinant DNA reagent | pET.32M.3C-hPINCH1 | This paper | LIM45(188-325)-/D295K/K297D/F253Y-H254N/ | BamHI/XhoI |
| Recombinant DNA reagent | pET.32M.3C-hPINCH2 | This paper | LIM5C(276-363) | BamHI/XhoI |
| Recombinant DNA reagent | pRSF-SUMO-hPINCH1 | This paper | (1-325) / Full-length | NdeI/XhoI |
| Recombinant DNA reagent | pETDuet-SUMO-hILK(C346S-C422S) / His-hParvinα | This paper | hILK(1-452)C346S-C422S / hParvinα(1-372) | (EcoRI/HindIII) for hILK (Recombinational method) for hParvinα |
| Peptide, recombinant protein | actin | Cytoskeleton, Inc | Cat. # AKL99 | Rabbit Skeletal Muscle |
| Strain, strain background (*Escherichia coli*) | BL21(DE3) | Kangti Health | Cat. # KTSM104L | |

*Continued on next page*

*Appendix 1—key resources table continued*

| Reagent type (species) or resource | Designation | Source or reference | Identifiers | Additional information |
|---|---|---|---|---|
| Strain, strain background (*Escherichia coli*) | Rosseta(DE3) | PMID:28966017 | | Dr. Mingjie Zhang (SUSTech, China) |
| Cell line (*Spodoptera frugiperda*) | IPLB-SF21-AE | Gibco/Thermo Fisher | Cat. # 11496015 | Maintained in Sf-900 II SFM, large scale in ESF 921 |
| Cell line (*Homo sapiens*) | HeLa | National Collection of Authenticated Cell Cultures | Cat. # TCHu187 | |
| Chemical compound, drug | Sf-900 II SFM | Gibco/Thermo Fisher | Cat. # 10-902-096 | Medium for *Sf9* cell |
| Chemical compound, drug | ESF 921 | Expression Systems | Cat. # 96-001-01 | Medium for *Sf9* cell |
| Chemical compound, drug | MEM | CORNING | Cat. # 10–010-CV | |
| Chemical compound, drug | DMEM | CORNING | Cat. # 10–013-CVRC | |
| Chemical compound, drug | Fetal Bovine Serum | PAN BIOTECH | Cat. # P30-3302 | |
| Chemical compound, drug | Fibronectin | Millipore | Cat. # FC020-5MG | |
| Chemical compound, drug | Cellfectin II Reagent | Thermo Fisher | Cat. # 10362100 | Transfection reagent for *Sf9* cell |
| Chemical compound, drug | Lipofectamine 2000 reagent | Invitrogen | Cat. # 11668–019 | Transfection reagent for HeLa cell |
| Chemical compound, drug | Lipofectamine 3000 reagent | Invitrogen | Cat. # L3000-015 | Transfection reagent for HeLa cell |
| Chemical compound, drug | Alexa Fluor 488 Phalloidin | Invitrogen/THermo Fisher | Cat. # A12379 | F-actin staining (1:100, v/v) |
| Chemical compound, drug | Alexa Fluor 594 Phalloidin | Invitrogen/THermo Fisher | Cat. # A12381 RRID: AB_2315633 | IF (1:200, v/v) |
| Chemical compound, drug | Alexa Fluor 647 Phalloidin | Invitrogen/THermo Fisher | Cat. # A22287 RRID: AB_2620155 | IF (1:100, v/v) |
| Chemical compound, drug | DAPI stain | SIGMA | D9542 | 1 µg/mL |
| Antibody | Anti-Paxillin (mouse monoclonal) | BD Bioscience | Cat. # 610620 RRID: AB_397952 | IF (1:500) |
| Antibody | Anti-Rsu1 (rabbit polyclonal) | Thermo Fisher Scientific | Cat. # A305-422A RRID: AB_2631813 | WB (1:4000) |

*Continued on next page*

*Appendix 1—key resources table continued*

| Reagent type (species) or resource | Designation | Source or reference | Identifiers | Additional information |
|---|---|---|---|---|
| Antibody | Anti-PINCH (mouse monoclonal) | BD Bioscience | Cat. # 612711 | WB (1:1000) |
| Antibody | Anti-Parvin (mouse) | Millipore | Cat. # MABT157 | WB (1:1000) |
| Antibody | Anti-ILK (mouse polyclonal) | BD Bioscience | Cat. # 611803 RRID: AB_399283 | WB (1:1000) |
| Antibody | Anti-GAPDH (mouse monoclonal) | TRANSGEN | Cat. # HC301-02 RRID: AB_2629434 | WB (1:3000) |
| Antibody | Anti-GFP (mouse monoclonal) | TRANSGEN | Cat. # HT801 | WB (1:3000) |
| Antibody | Anti-Mouse IgG (H+L), Alexa Fluor 594 (donkey) | Thermo Fisher Scientific | Cat. # A21203; RRID: AB_141633 | IF (1:1000) |
| Antibody | Anti-mouse IgG, HRP-linked Antibody (horse) | Cell Signaling | Cat. # 7076 RRID: AB_330924 | WB (1:10000) |
| Antibody | Anti-rabbit IgG, HRP-linked Antibody (goat) | Cell Signaling | Cat. # 7074 RRID: AB_2099233 | WB (1:10000) |
| Commercial assay or kit | Western ECL substrate | BIO–RAD | Cat. # 170–5061 | |
| Software, algorithm | ASTRA6 | WYATT Technology | PRID:SCR_016255 | |
| Software, algorithm | MicroCal PEAQ-ITC integrated Software package | Malvern Panalytical | | |
| Software, algorithm | OriginPro | OriginLab | Learning Edition | |
| Software, algorithm | GraphPad Prism | GraphPad Software | RRID: SCR_002798 | |
| Software, algorithm | Image pro plus | MEDIA CYBERNETICS | RRID: SCR_016879 | |
| Software, algorithm | Image J | National Institutes of Health | RRID: SCR_003070 | |

