## [Decision Letter]

**Acceptance summary:**

RSU1 is a member of a quaternary protein complex, which consists of RSU1, a pseudokinase (ILK) and two F-actin binding proteins (PINCH and Parvin). The complex plays fundamental roles during development, postnatal homeostasis and disease. RSU1 is the least investigated member of this protein complex, and has been postulated to modulate actin dynamics. The paper by Yang et al. sheds light on the mechanism of this elusive RSU1 function. The finding reported by Yang and co-workers will have a significant impact on the adhesion field.

**Decision letter after peer review:**

Thank you for submitting your article "Complex structures of Rsu1 and PINCH1 reveal a regulatory mechanism of the ILK/PINCH/Parvin complex for F-actin dynamics" for consideration by *eLife*. Your article has been reviewed by three peer reviewers, including Reinhard Fässler as the Reviewing Editor and Reviewer #1, and the evaluation has been overseen by John Kuriyan as the Senior Editor. The following individual involved in review of your submission has agreed to reveal their identity: Ben Goult (Reviewer #3).

The reviewers have discussed the reviews with one another and the Reviewing Editor has drafted this decision to help you prepare a revised submission.

Summary:

RSU1 is a member of a quaternary protein complex, which consists of RSU1, a pseudokinase (ILK) and two F-actin binding proteins (PINCH and Parvin). The complex plays fundamental roles during development, postnatal homeostasis and disease. RSU1 is the least investigated member of this protein complex. It is best known for controlling Ras GTPase activity. While RSU1 has been postulated to modulate actin dynamics, the underlying mechanisms were never seriously investigated. The paper by Yang et al. sheds light on the mechanism, which will have a significant impact on the adhesion field.

Essential revisions:

1) The quality of the actin bundling assays is not convincing and the assays therefore require considerable improvement. Specifically, the number and size of bundles should be quantified. An example of acceptable in vitro actin assembly/bundling assays can be found, for example, in EMBO J (2008)27:2943-2954.

2) The effects of RSU1 mutations were studied by overexpressing RSU1 and its mutants in cultured cells. Since the authors themselves in the Discussion section that "It is likely that cells tightly control the Rsu1 level for the homeostasis of actin dynamics, […] would interfere with the actin-dependent cell motility", it is very important to perform the analyses with "rescue experiments" in RNAi-depleted or CRISPR/Cas9-mediated knockout cells.

3) The authors used only RSU1 and PINCH1 mutants (RSU1 Y140K or D115K/Y140K, and PINCH1 K297D or D395K) that disrupt the polar interactions shown in Figure 3D. The authors identified 22 surface residues in RSU1 that formed H-bonds, salt bridges, and hydrophobic interactions, and the C-tail of PINCH1 (=WH2 motif) for binding to RSU1. We propose to extend the binding studies with mutants that inhibit the binding of RSU1 to PINCH1 to better demonstrate the mechanism of complex formation more strongly.

4) The quality of the HeLa immunostaining is not convincing and should also be improved. Particularly the high background (Figure 5) is problematic, FAs and stress fibers are not well visible. Also, the GFP-RSU1 should be in FAs but looks uniformly distributed, similarly like the RSU1 mutants., FAs and stress fibers are not well visible. The authors should show that wild type and mutant RSU1 are expressed at similar/comparable levels.

5) RSU1 was shown to bind PINCH1 but not PINCH2. The binding of PINCH2 to RSU1 reported in the present manuscript should therefore, be confirmed with an orthogonal assay (e.g. MST).

---

## [Author Response]

Essential revisions:1) The quality of the actin bundling assays is not convincing and the assays therefore require considerable improvement. Specifically, the number and size of bundles should be quantified. An example of acceptable in vitro actin assembly/bundling assays can be found, for example, in EMBO J (2008)27:2943-2954.

As suggested, we quantified the number and length of actin bundles in different samples. Consistent with our observation in Figure 4D, both the number and length of the actin bundles was significantly decreased when the IPP complex was mixed with Rsu1. On the other hand, binding-deficient mutants of Rsu1 showed little or no effect on the IPP-induced actin bundling. Adding Rsu1-WH2 led to highly crowded bundles, which is difficult for quantitative analysis. We have included this part of data as Figure 4—figure supplement 2 and added the related description in the section of “The PINCH1/Rsu1 interaction blocks the F-actin bundle formation promoted by the IPP complex”.

In addition, we analyzed the actin bundle formation using negative-staining samples under electron microscope. In line with our light microscopic analysis, we observed large actin bundles in the F-actin sample with the IPP complex, while further adding Rsu1 to the sample eliminated large actin bundles. We also included this as Figure 4—figure supplement 3 in the revised manuscript.

2) The effects of RSU1 mutations were studied by overexpressing RSU1 and its mutants in cultured cells. Since the authors themselves in the Discussion section that "It is likely that cells tightly control the Rsu1 level for the homeostasis of actin dynamics, […] would interfere with the actin-dependent cell motility", it is very important to perform the analyses with "rescue experiments" in RNAi-depleted or CRISPR/Cas9-mediated knockout cells.

We may not address our point clearly here. Actually, previous studies conducted by Kadrmas, et al., 2004 and Gonzalez-Nieves, et al., 2013, have showed that in either *Drosophila* or mammalian cells, the depletion of Rsu1 led to a decreased protein level of PINCH1, which may interfere with the IPP complex formation and thereby impair actin cytoskeleton regulation. We have cited these papers in the Discussion part. During revision, we also prepared Rsu1-depleted HeLa cells by using RNAi technique and used the Rsu1-knock out (KO) HT1080 cell provided by Prof. Yi Deng (Wang et al., 2020 JBC, DOI: 10.1074/jbc.RA120.014413). Consistent with the previous findings, the cell morphology and stress fiber formation were largely disrupted in the Rsu1-depleted cells and the re-expression of wild-type Rsu1 but not the Y140K nor DY/KK mutants could partially rescue the decreased stress fiber formation, presumably due to the decreased level of the IPP components (see Author response image 1 and 2). Considering that the data did not provide additional insights compared with the previous studies and our paper mainly focus on the inhibition role of Rsu1 on actin bundling function of IPP, we decided not to include the data in the revised manuscript.

**Author response image 1. sa2fig1:** Depletion of Rsu1 in HeLa cells and rescue effects of wild type Rsu1 and its mutants. (A) Confocal images of HeLa cells transfected with control siRNA and GFP-vector, or Rsu1-siRNA with GFP-vector, Rsu1-GFP or Rsu1 mutants. FAs and stress fibers were stained by paxillin and phalloidin, respectively. (B) The proteins level in normal cells or cells transfected with different siRNA or constructs was detected by western blot. The Rsu1-siRNA efficiently knocked down the endogenous Rsu1 and the GFP-Rsu1 or the mutants was expressed at a similar protein level, which are also comparable with the endogenous Rsu1 expression level. The protein level of ILK and α-parvin were also checked by western blot. (C) Quantification of the percentage of HeLa cells containing normal stress fiber formation as shown in a. Data was collected from 10 images in each sample. In each image, more than 10 cells were quantified. Significance was calculated by using student t test in GraphPad Prism. ***p<0.001, *p<0.05.

**Author response image 2. sa2fig2:** Rescue effect of wild type Rsu1 and its mutants in Rsu1-KO HT1080 cells. (A) Confocal images of normal HT1080 cells or Rsu1-KO cells transfected with GFP-vector, Rsu1-GFP or Rsu1 mutants. FAs and stress fibers were stained by paxillin and phalloidin, respectively. (B) The proteins level of each cell in panel A was checked by western blot.

3) The authors used only RSU1 and PINCH1 mutants (RSU1 Y140K or D115K/Y140K, and PINCH1 K297D or D395K) that disrupt the polar interactions shown in Figure 3D. The authors identified 22 surface residues in RSU1 that formed H-bonds, salt bridges, and hydrophobic interactions, and the C-tail of PINCH1 (=WH2 motif) for binding to RSU1. We propose to extend the binding studies with mutants that inhibit the binding of RSU1 to PINCH1 to better demonstrate the mechanism of complex formation more strongly.

As suggested, we designed seven more interface mutations on Rsu1 and measured their effect on the binding affinity by using ITC. Consistent with our structural analysis, these mutations either weakened or abolished the Rsu1/PINCH1 interaction. We have included this as Figure 3—figure supplement 2 in the revised manuscript.

4) The quality of the HeLa immunostaining is not convincing and should also be improved. Particularly the high background (Figure 5) is problematic, FAs and stress fibers are not well visible. Also, the GFP-RSU1 should be in FAs but looks uniformly distributed, similarly like the RSU1 mutants., FAs and stress fibers are not well visible. The authors should show that wild type and mutant RSU1 are expressed at similar/comparable levels.

We agree with the reviewers that the quality of the cell imaging was not high enough. During revision, we redid the cellular experiments to improve the image quality. In these experiments, we followed the suggestion to sort the cells by using flow cytometry. The cells with comparable GFP-signal were collected for the following imaging analyses. The expression levels of Rsu1 and its mutants were further checked by western blot. The updated images in Figure 5A, 5B, and Figure 6D showed the results with a higher quality. The western blot result was added in the manuscript as Figure 5—figure supplement 1.

5) RSU1 was shown to bind PINCH1 but not PINCH2. The binding of PINCH2 to RSU1 reported in the present manuscript should therefore, be confirmed with an orthogonal assay (e.g. MST).

We thanks the reviewers for their valuable suggestion. We used ITC to measure the binding affinity between PINCH2_LIM5C and Rsu1. Interestingly, we found that the binding of PINCH2_LIM5C to Rsu1 (K_d_ of ~2.0 μM) was much weaker than the binding of PINCH1_LIM5C to Rsu1 (K_d_ of ~15 nM). The sequence alignment of PINCH proteins (Figure 1—figure supplement 2) showed that F253 and H254 in PINCH1 as the interface residues were replaced by a Tyr and an Asn in PINCH2, respectively. Presumably, the substitution of F253 to a tyrosine weakens the hydrophobic interaction by adding a hydroxyl group and the substitution of H254 to an asparagine disrupts the pi-pi staking between H254_PINCH1_ and H90_Rsu1_. We have included this part of data in Figure 3, Figure 1—figure supplement 2 and Figure 1—figure supplement 4. In addition, we designed the F253Y/H254N mutation in PINCH1 to check whether this mutation blocks the binding of PINCH1_LIM45C to Rsu1. Unfortunately, the protein quality of the F253Y/H254N mutant was not good enough for ITC analysis (Author response image 3).

**Author response image 3. sa2fig3:** SDS-PAGE analysis of the purified PINCH proteins.